# Influence of Practitioner-Related Placement Variables on the Compressive Properties of Bulk-Fill Composite Resins—An In Vitro Clinical Simulation Study

**DOI:** 10.3390/ma15124305

**Published:** 2022-06-17

**Authors:** Tamar Brosh, Moshe Davidovitch, Avi Berg, Aviran Shenhav, Raphael Pilo, Shlomo Matalon

**Affiliations:** 1The Department of Oral Biology, The Maurice and Gabriela Goldschleger School of Dental Medicine, Sackler Faculty of Medicine, Tel Aviv University, Tel Aviv 6997801, Israel; rafipilo@tauex.tau.ac.il; 2Department of Orthodontics, The Maurice and Gabriela Goldschleger School of Dental Medicine, Sackler Faculty of Medicine, Tel Aviv University, Ramat Aviv, Tel Aviv 6997801, Israel; davidom@tauex.tau.ac.il; 3Department of Oral Rehabilitation, The Goldschleger School of Dental Medicine, Sackler Faculty of Medicine, Tel Aviv 6997801, Israel; aviberg@mai.tau.ac.il (A.B.); aviranshenhav@gmail.com (A.S.); matalons@tauex.tau.ac.il (S.M.)

**Keywords:** bulk-fill, practitioner, compressive, oral quadrant

## Abstract

Aims: To determine if restoration location and/or execution behavior force parameters have an influence on the mechanical properties of bulk-fill composite dental restorations. Methods: Pressure transducers were placed within each quadrant of dental mannequin jaws. Cylindrical molds were placed above the transducers and filled with two bulk-fill composite materials, Filtek and Tetric, by four experienced dentists. Each dentist prepared five specimens per quadrant and material. The total placement time, mean force, number of peak forces (above 25 N), and mean peak(s) force during placement were measured. Then, the stiffness and maximal compressive strength of the specimens were determined while loading the specimens up to failure using a universal loading machine. Results: Placement time was affected by jaw (*p* < 0.004) and side (*p* < 0.029), with the shortest time demonstrated for the left side of the mandible. Force exerted during restoration placement was not normally distributed without differences in location (jaw) or material. A higher application force was found on the right side (*p* < 0.01). The number of peak forces was affected by side (*p* < 0.03), with less peaks on the left side. No significant differences were found in compressive strength when correlated to restoration location, participant, or material (*p* = 0.431). The stiffness values of Filtek (3729 ± 228 N/mm) were found to be 15% higher than Tetric (3248 ± 227 N/mm) (*p* < 0.005). No correlations were found between the compressive strength or stiffness and the amount of force applied during placement. Conclusions: The individual restoration material placement parameters did influence practitioner performance; however, these differences did not affect the mechanical properties of the final restoration.

## 1. Introduction

Advances in material sciences and treatment techniques in dentistry have concomitantly elevated esthetic demands within the average patient population of modern Western societies [1,2]. This emphasis on esthetic outcomes has perhaps found its greatest expression in the increased use of resin composite restorative materials. This has driven developers to improve the mechanical properties of these materials to parallel esthetic demands [3].

Dental composites are classified according to their matrix and filler types, for which composition and quantity determine their physical and mechanical properties. Moreover, their viscosity and flow characteristics have been shown to affect their behavior, which influences their handling and further impacts their mechanical properties [3,4]. Obviously, the prognosis of any such restoration depends not only on its mechanical properties, but also on its physical properties, such as volumetric shrinkage and cavity configuration (C-factor) [5]. To prevent the restoration from detaching from the prepared tooth walls, the bond strength of the bonding material to both the tooth and the composite resin should be equivalent to or greater than those of the stress developed due to the contraction of the material during polymerization [6].

Practitioner influence on the longevity of composite restorations as reflected by annual restoration failure rates has been previously reported; however, this has been reported as being a determining [7,8,9,10,11] and non-determining factor [12]. It has also been reported that the intra-oral location of restoration placement may also influence its longevity, but these studies are few and inconclusive [13,14,15]. For example, Pallesen et al. showed that the survival of posterior restorations in the upper jaw was higher compared with the lower jaw [13], whereas Lucarotti et al. reported the opposite [14]. Furthermore, a meta-analysis of clinical studies did not provide data regarding differences in restoration longevity between the upper and lower jaw [15]. It should be noted that in all these studies, the association between the sidedness of the restoration as related to longevity was not investigated. It is worthy of consideration that the influences potentially imposed by this, as well as of which jaw is restored, may each or together affect restoration longevity.

The mechanical properties of restorative materials are commonly tested on specimens prepared on a laboratory bench without considering the location of the restoration in the oral cavity [16,17,18,19]. In vitro laboratory studies have shown differences in the porosity of composite resin specimens created by different practitioners [20]. These have been reported to influence the mechanical properties of restorations [21]. These differences in porosity were also found when specimens prepared outside the oral cavity were compared with simulated oral cavity specimens [22].

Current advances in polymerization chemistry have resulted in materials that have been reported to have improved physical properties and shortened working times [23]. These materials allow for insertion of up to 4–6 mm of composite material thickness, without compromising their mechanical properties, aptly labeling these materials as “bulk-fill”. In addition, these are also claimed to have a reduced shrinkage with improved structural composition and higher abrasion resistance [24]. Lower shrinkage stresses were analyzed by finite element models of bulk-fill flowable restorations compared with conventional resins [25]. Both Class I and Class II posterior restorations have shown significant stress concentrations due to shrinkage [26]. It has also been reported that the comparison of these to conventional composite resins demonstrates comparable degrees of conversion [27], a lower modulus of elasticity [28], a less detrimental influence of light source orientation [29], and reduced shrinkage at curing [30,31]. These benefits are achieved through several approaches: more translucent materials, as well as a lower filler concentration allowing greater light penetration and more efficient photo activation [32]. Moreover, secondary caries are of great concern, especially in Class II restorations. New developments in bioactive composites with antibacterial properties might improve their clinical performances [33].

The ergonomic and behavioral advantages of these materials have led to an increase in their clinical acceptance; however, manufacturer recommendations regarding use are not universal. Specifically, these have been provided as follows: either that only the 2 mm most occlusal area of the cavity is restored with a higher filled composite resin for improved durability and resistance to abrasion, or that this final layer and the underlying mass of the restoration all be comprised of the same bulk-fill composite resin material. Therefore, further investigation about the clinical handling of bulk-filled composite resins is needed to better understand its influence on the mechanical properties of this material. Furthermore, the properties of these materials might also be influenced by the intraoral location of the cavity and related accessibility for preparation. The hypothesis was that the restoration placement location in the oral cavity may influence the mechanical properties of such restorations. The aims of this in vitro study are (1) to characterize several practitioner-related force parameters exerted during the placement of the restoration and (2) to determine if these parameters and the location of the restoration have an influence on its compressive strength.

## 2. Materials and Methods

### 2.1. The In Vitro Set-Up

Dental mannequin jaws were equipped with model CSMN-5L miniature pressure transducers with digital force display R320 (Larit, Ra’anana, Israel), located at the first molar teeth in each quadrant (Figure 1). Cylindrical Teflon molds (3 mm internal diameter, 8 mm in length) were located above the transducer to simulate the conditions of Class I cavity preparations. The molds were required to be 2 mm longer than the final specimen because the apical 2 mm encircled the extruded part of the transducer, yielding a final specimen with dimensions of 3 × 6 mm. The mannequin jaws were placed in a phantom-head in order to simulate the patient’s position during restorative treatment. The participants were seated in their preferred corresponding treatment manner.

The specimens were prepared in one increment from two bulk-fill materials: Filtek Bulk-Fill (3M/ESPE, St. Paul, MN, USA) and Tetric Bulk-fill (Ivoclar-Vivadent, Schaan, Liechtenstein). The forces expressed by each participant during the placement of the dental restorative composite in the simulated Class I cavity preparations were measured in Newtons. Data were monitored as force (N) vs. time in seconds (s) (Figure 2). Curing (20 s) followed material placement (ART L5 Curing Light, BONART CO., New Taipei, Taiwan). Ambient room lighting was the only source of illumination during this trial (i.e., without a unit lamp).

Four experienced (each with more than 15 years of general clinical practice) right-handed dentists (two males and females) participated in the study. Each dentist prepared five specimens in each of the four quadrants from each material, yielding 20 specimens per material. A uniform armamentarium was utilized by each participant; specifically, a resin composite plugger (Super Plugger Medium/Large, American Eagle, Missoula, MT, USA), dental spatula (Composite 3, American Eagle, Missoula, MT, USA), and 8 mm straight celluloid matrix transparent strips (Black Bird Ltd., London, UK) and a dental mirror.

### 2.2. On-Line Placement Measurements and Analysis

The characteristic force−time curves acquired during placement were plotted (Figure 2). Four parameters were analyzed from the data of each specimen preparation: (1) total placement time, (2) mean force exerted during placement, (3) number of peak forces (i.e., above 25 N), and (4) mean peak(s) force.

The specimens were extracted from their molds and isolated from any light source for two weeks when mechanical testing was performed.

### 2.3. Compressive Test

The dimensions of each specimen were measured using a digital caliper from which the cross-sectional area was calculated. These were then tested in compression using a universal testing machine (Instron, model 4500, Buckinghamshire, UK) equipped with a 10 kN load-cell with a cross-head speed of 0.5 mm/min and compression plates. The stiffness of each specimen was calculated from the linear slope of the resultant force curve. The maximal compressive force at failure was considered and derived from the measured force data by dividing the maximal compressive force by the cross-sectional area.

### 2.4. Statistics

The mixed model analysis (IBM, SPSS Ver. 27, Armonk, NY, USA) was applied to find the differences between restoration placement parameters and mechanical properties. The independent variables were the practitioner, restoration location, and material. Statistical significance was determined for *p* < 0.05.

## 3. Results

### 3.1. Placement Parameters

Figure 3 is a representative diagram obtained during restoration placement by all practitioners who participated in the study using a Filtek material at the upper left quadrant. It can be seen that the duration of placement, as well as the pattern of force application varied according to each of these variables. It was found that in some instances, peak forces were evident (large or small), while in others, almost no peaks could be detected.

Figure 4 presents the time needed for restoration placement in either jaw, as related to practitioner and material (Figure 4a,b). It was found that the jaw (*p* < 0.004) and side (*p* < 0.029) significantly influenced the time needed for placement (Figure 5a,b). The shortest time monitored was in the left side of the mandible. However, the choice of material was not found to affect the time required for placement (*p* = 0.95).

The mean force exerted during restoration placement (Figure 6) was found not to be normally distributed, as determined by plotting the raw data; therefore, the analysis was performed on log values. The mean force applied during restoration on the right side (73.23 ± 79.9 N) was greater than that found on the left side (55.24 ± 53.1 N) (*p* < 0.01). However, the choice of jaw and/or material did not influence the mean force (*p* > 0.495).

The number of peaks (Figure 7) were affected by the side (*p* < 0.03). Specifically, less peaks were found on the left side. The interaction between jaw and side was significant (*p* < 0.016), for example, more peaks and a higher mean peak force were found on the left side of the mandible.

### 3.2. Compressive Properties

No significant differences were found in compressive strength as affected by the restoration location, participant, or material (*p* = 0.431). The stiffness values of Filtek (3729 ± 228 N/mm) were found to be 15% higher compared with Tetric (3248 ± 227 N/mm) (*p* < 0.005). No correlations were found between the compressive strength or stiffness and the amount of force applied during placement.

## 4. Discussion

The outcome of this study showed differences in restoration placement parameters between different practitioners and differences between the jaws and sides; however, these differences did not affect the final compressive strength of the bulk-fill resins tested in this study, thus rejecting the hypothesis. The variance in stiffness between the materials could be attributed to the differences in their material compositions.

The less time needed for restoration placement in the left side of the mandible is to be expected because this quadrant is the most directly accessible for the right-handed practitioner.

Interestingly, the number of peaks or the force applied during placement did not affect the in vitro compressive properties. The placement technique and material manipulation directly influence restoration porosity, as this relates to trapping the air within the restoration. This may influence its longevity, as it is related to the cyclic loading applied during mastication. The present study did not include the aging of specimens before static mechanical testing, but did not find differences between the compressive properties prepared using different manipulations. Furthermore, it was found that porosity did not affect the fracture resistance of bulk-filled specimens [34].

The relationship between practitioner influence and the failure rate of the composite restorations has been reported to be both dependent [7,8,10,35] and non-dependent [12]. The present study found no differences in restoration mechanical properties between the sides (left/right quadrants) and jaws (maxillary/mandibular). These findings differ from those of Laske et al. [8,9] and those of Locarotti et al., where these factors were found to influence restoration longevity [14]. Collares concluded that the operator is associated with restoration longevity, with a higher risk of failure reported for the maxillary anterior dentition than the mandibular [11]. According to Huang, restoration longevity and failure risk are highly dependent on bonding between the restoration and the tissue, and that the bonding quality is practitioner related [36]. This aspect was not considered in this study. However, it has also been reported that, ultimately, restoration failures occur mostly related to secondary caries and restoration fracture, but that restoration survival is unaffected by the restored surfaces [37].

The influence of the practitioner on the longevity of restorations was studied by Stewardson et al. [38]. They reported that practitioners significantly affected the survival of Class V bulk-fill restorations. Based on some non-significant differences between the jaws and sides, they assumed that the location of the restoration might affect the observed failures. However, the stresses developed on Class V restorations were not pure compressive, as analyzed in the present study, and other factors such as the recurrent caries preparation technique may influence these findings [38].

Another review considering different restorative materials for Class I and II restorations concluded that restoration failure was also related to practitioner influence [39]. In addition, it has been reported that the survival of posterior restorations in the upper jaw was higher than in the lower jaw [13,40], and, conversely, that this was true of the mandibular arch [14].

Studies reporting on the time required for restoration placement are very limited. It has been shown that the time required to perform dental restorations with bulk-fill materials is shorter compared with conventional resins, without considering the location of the restoration in the oral cavity [23,41]. Other techniques for quick posterior restorations can be considered [42]. The present study found that differences in placement times between the jaws and sides were a few seconds, which is attributable to practitioner comfort. However, as mirror-assisted placement required more time, as was found in this study, it can be concluded that the total time for Class I restoration is longer when direct visualization and when access is not available. Moreover, the incorporation of air bubbles related to increased material operator manipulation as related to placement forces or peak forces may be more significant variables than the time taken to compose a given composite restoration.

The main limitation of this study is the relatively small number of participants. However, this is somewhat mitigated by the application of pressure sensors that could identify the practitioner’s parameters. This study considered the intrinsic compressive properties of the tested materials, and further studies ongoing in our laboratory will consider the influence of the placement procedure of bulk-filled materials, including the aging of specimens and bonding to extracted teeth in the four quadrants of the mouth. In such studies, the shrinkage of the materials will not be eliminated, because it might influence the interface between the restoration and the dentin. Moreover, it would be interesting to compare the placement parameters of left-handed practitioners.

## 5. Conclusions

To conclude, the clinical simulation setting in which the behavior of bulk-fill composite resin restoration materials was tested indicates that operator-related factors do influence performance; however, these were not found to affect the physical properties of the completed restorations, as related to the compressive properties. Further research is needed to test the influence of operator-related factors on other physical properties of the completed restorations, such as the tensile properties.

## Figures and Tables

**Figure 1 materials-15-04305-f001:**
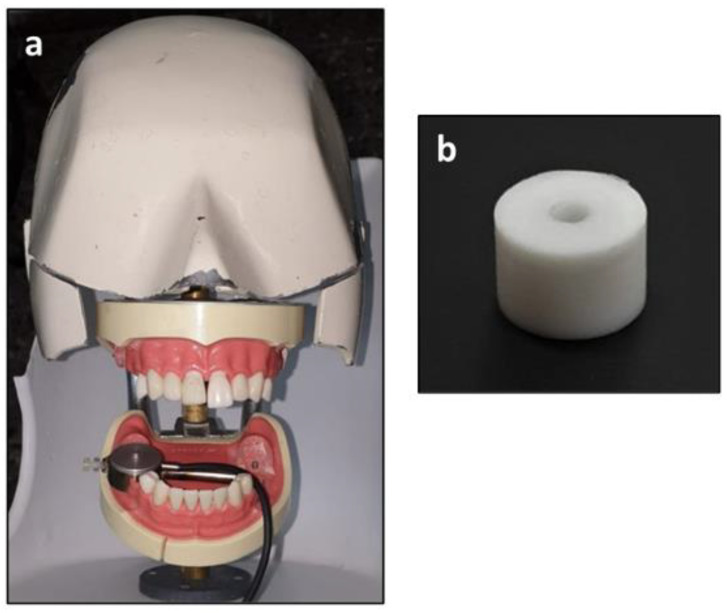
The mannequin setup. (**a**) The transducer is located at the molar area of the lower right quadrant. (**b**) the mold for preparing the specimens.

**Figure 2 materials-15-04305-f002:**
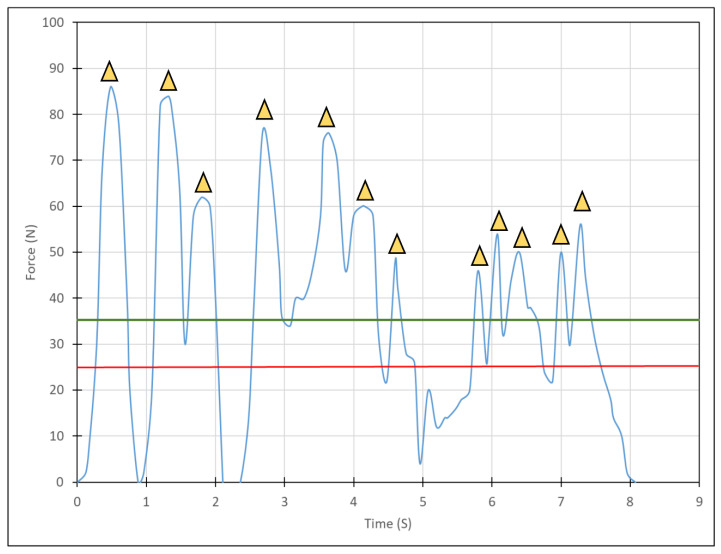
A characteristic force-time diagram acquired during restoration placement. The yellow triangles represent the local peak forces during the procedure where the peaks were considered as above the minimum values of 25 Newtons (red solid line). The mean force expressed by the practitioner during restorative material placement is also presented (green solid line).

**Figure 3 materials-15-04305-f003:**
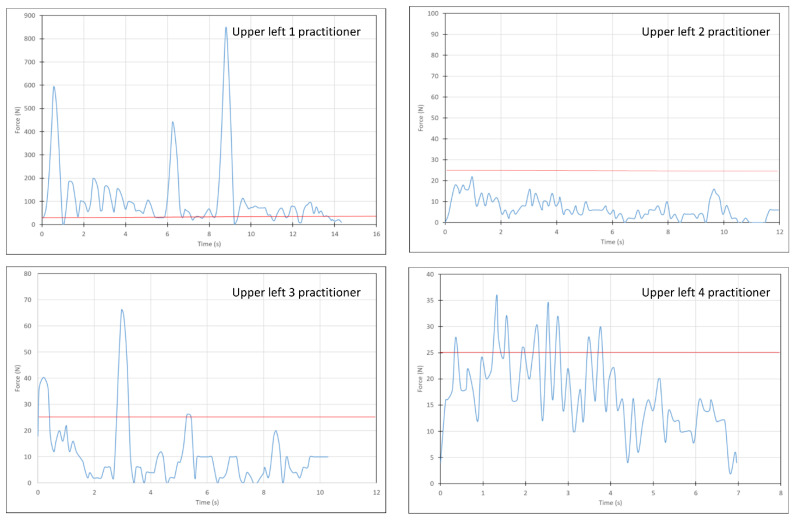
Characteristic force-time diagrams of Filtek specimen preparation in the upper left quadrant obtained from each practitioner. The local peak forces during the procedure were considered as above the minimum values of 25 Newtons (red solid line, see Figure 2).

**Figure 4 materials-15-04305-f004:**
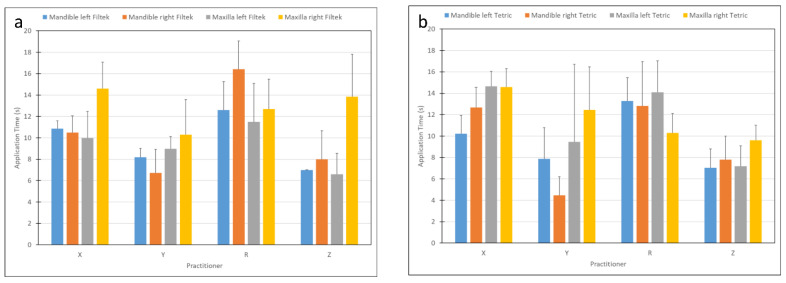
Restoration placement application time as related to practitioner and materials: (**a**) Filtek, (**b**) Tetric.

**Figure 5 materials-15-04305-f005:**
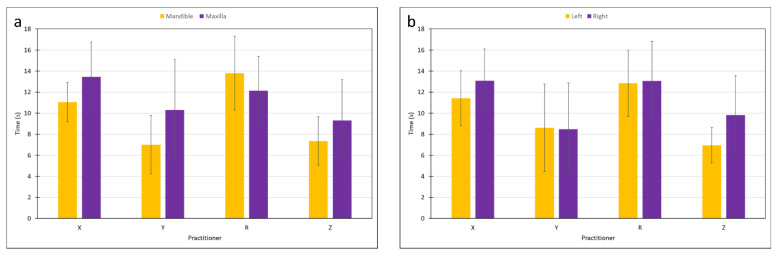
Restoration placement application time as related to practitioner and (**a**) jaw, (**b**) side.

**Figure 6 materials-15-04305-f006:**
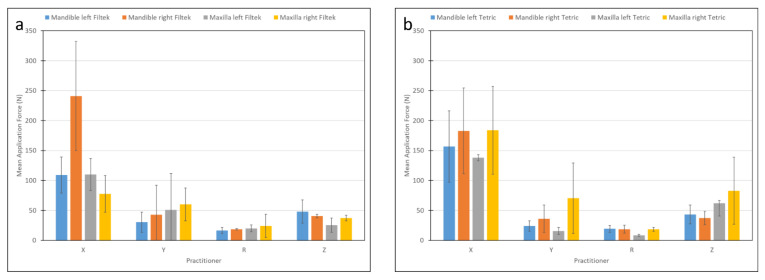
Mean application force as related to practitioner and materials: (**a**) Filtek, (**b**) Tetric.

**Figure 7 materials-15-04305-f007:**
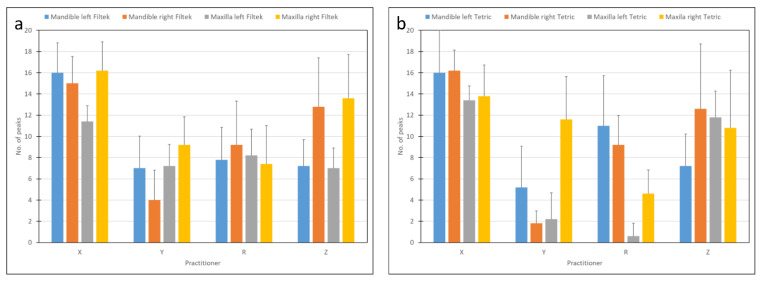
Number of peaks as related to practitioner and materials: (**a**) Filtek, (**b**) Tetric.

## Data Availability

Data can be obtained from the corresponding author.

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
