# Peer review of "Influence of Practitioner-Related Placement Variables on the Compressive Properties of Bulk-Fill Composite Resins—An In Vitro Clinical Simulation Study"

_materials, 2022, doi:10.3390/ma15124305_

Round 1
Reviewer 1 Report
The present study aimed to determine if restoration location and/or execution behavior force parameters have an influence on the mechanical properties of bulk-fill composite dental restorations. The subject is interesting, however the text requires significant improvements and the final decision depends of the authors satisfactory corrections:
Template:
The journal guidelines and template should be corrected.
Title:
Please describe the study type in your title.
Abstract:
Provide the sample size and groups division;
Who applied the resin? Students? Dentists?
“stiffness and maximal compressive strength tolerated to restoration failure were determined.” How?
Introduction:
“It has also been reported that the intra-oral location of restoration placement may also influence its longevity but these studies are few and inconclusive” Please add the references.
According to your text, a meta-analysis of clinical studies showed no differences in the longevity of restoration between the upper and lower jaw. So, why it is worth to develop an in vitro study with this subject? What is missing from the meta-analysis data that would be complemented by your study?
Describe the effect of shrinkage stresses generated by the polymerization shrinkage of buk-fill and resin composites (https://doi.org/10.3390/ma14092366 and https://doi.org/10.3390/app11052215).
What are your study’s hypothesis?
Methods:
“The participants were seated in their preferred corresponding treatment manner.”” Who are the participants? How many participants? How they were standardized, selected or excluded?
Does the participants access to dental mirror? How they are able to restore the upper teeth if this tool is not listed?
Is the number of participants enough to elucidate this study outcome? How it was rationally determined?
What is your sample size? How it was calculated?
What was the time to finish the procedure? How the ambient light was considered to not modify the resin property?
What is the typodont material? Is it bond strength with resin composite comparable to human tissue? Otherwise, your fracture load test is considering a setup different from the in vivo condition.
How the compressive load was applied? What kind of antagonist was used? And about the contact points with it?
Describe the normality test.
Discussion:
Provide a paragraph with all your study’s limitations.
Author Response
Please see the attachment for reviewer 1

Reviewer 2 Report
The article raises a very interesting issue.
The abstract illustrates the work correctly. However, there should be no words describing what part of the abstract it is, such as introduction, conclusions, etc.
There are no keywords.
The introduction illustrates the theoretical issues raised in the manuscript very well; however, it is quite long. I suggest shortening it and adding some information for discussion.
Please provide a null hypothesis.
Please provide the bioethics committee approval number.
Have sample size calculations been performed? On what basis was the number of samples determined?
Results and discussion are sufficiently and correctly discussed.
There is a lack of description of the study's strengths and limitations. Please describe further perspectives on the development of the study.
Conclusions answer the research question posed. However, they should be in a separate subsection.
Author Response
Please see the attachment to reviewer 2

Reviewer 3 Report
The manuscript is interesting and the study design is interesting and well structured.
I suggest adding a few citations regarding microleakage in class II bulk-fill restorations and considering the evaluation of the effect that novel bioactive restorative composites could have in the survival and success of composite restorations.
conclusions seems pretty shallow, I would like you to further expand on future objective from your research team regarding this topic: for example, performing ageing of the samples, comparing it with bioactive composite resins (perhaps evaluating the antibacterial capability over time, in vitro, using bacterial coltures) and comparing left handed and right handed operators could be interesting follow-up studies.
Author Response
Please see the attachment for reviewer 3

Round 2
Reviewer 1 Report
The reviews were carried out satisfactorily.
Author Response
Thank you for your suggestions.
Reviewer 2 Report
The authors responded to all reviewer comments. In my opinion, the article in its current form can be published in MDPI Materials.
Author Response
Thank you for your suggestions.